# Adaptive Sample-space & Adaptive Probability coding: a neural-network based approach for compression

## Abstract

We propose Adaptive Sample-space & Adaptive Probability (ASAP) coding, an efficient neural-network based method for lossy data compression. Our ASAP coding distinguishes itself from the conventional methods based on adaptive arithmetic coding in that it models the probability distribution for the quantization process in such a way that one can conduct back-propagation for the quantization width that determines the support of the distribution. Our ASAP also trains the model with a novel, hyper-parameter free multiplicative loss for the rate-distortion tradeoff. With our ASAP encoder, we are able to compress the image files in the Kodak dataset to as low as one fifth the size of the JPEG-compressed image without compromising their visual quality, and achieved the state-of-the-art result in terms of MS-SSIM based rate-distortion tradeoff.

## 1 Introduction

In terse terms, *lossy data compression* is a task in which one seeks to encode a data file into as short a code as possible without losing the essential information. Extensive research has been conducted in the field of lossy data compression; JPEG, WebP and BPG are well known lossy image compression codec. The recent advances in machine learning methods are accelerating the pace of the research in this field. For example, studies like Toderici et al. (2015; 2017); Ballé et al. (2017); Theis et al. (2017); Johnston et al. (2017); Rippel & Bourdev (2017); Mentzer et al. (2018); Agustsson et al. (2018); Nakanishi et al. (2018); Ballé et al. (2018) have incorporated the methods of deep learning into lossy compression, and some of them succeeded in producing a result that far surpasses the classical, neural-network-free methods.

Almost all lossy codecs to date—including the state-of-the-art codec—are built on autoencoder. After transforming the input data by the autoencoder, the lossy codecs discretize the latent space using a *quantizer* function and further converts the transformed data so that it can be stored as a binary code. In order to quantize the latent feature into as short a code as possible, one usually decompose the feature into pieces and use the information of the pieces that it has quantized so far (i.e., context) in order to adaptively choose a discrete probability space for the quantization and the entropy coding of the next piece of information. By construction, the choice of the probability space for the quantization and the entropy coding of the feature vector is a key factor that determines the performance of the compression algorithm. In this study, we present a novel compression architecture that allows the user to optimize the discrete support of *the probability space to be used for quantization and entropy coding* (PSQE). Optimization of the support of the PSQE in a NN-based architecture is a daunting task. To the authors' best knowledge, there has been no study to date that enabled this optimization for NN-based architecture. Our Adaptive Sample-space & Adaptive Probability (ASAP) coding partially achieves this task by assigning to each latent feature a discretized normal distribution with different mean, variance and quantization width. Our method automatically chooses appropriate quantization width for each latent feature. For the compression with small models in particular, we were able to confirm that this adaptive quantization width benefits the performance in terms of the MS-SSIM score of the reconstructed images.

We also present a novel objective function for the training of the quantizer. In general, the problem of minimizing the reconstruction error ($\mathtt{distortion}$) under a constraint of code-length (or $\mathtt{bpp}$) can be

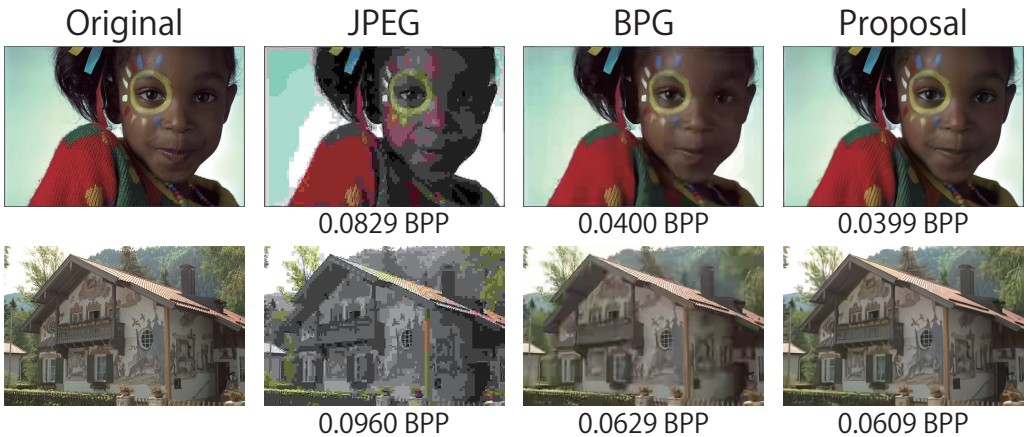

Figure 1: Reconstructed images with different compression methods.

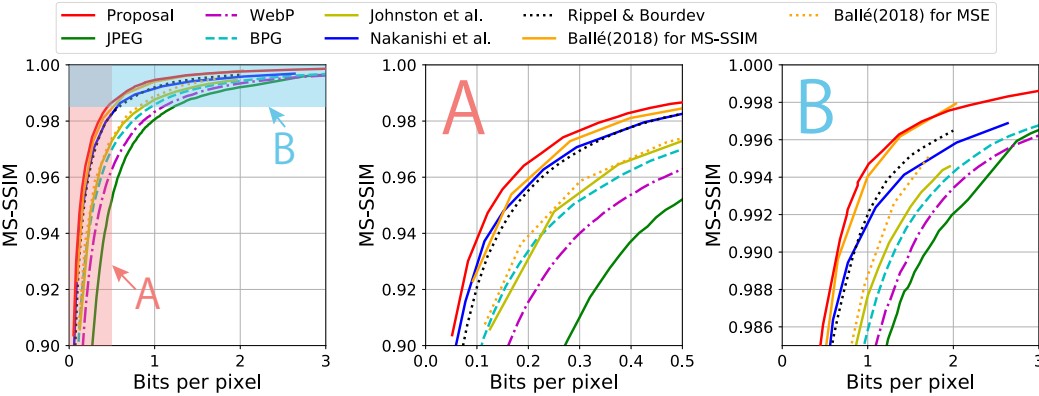

Figure 2: Rate-distortion tradeoff curves evaluated for different methods on the Kodak dataset. The horizontal axis represents bits-per-pixel (bpp) and the vertical axis represents the average multi-scale structural similarity (MS-SSIM) computed over RGB channels. The right two panels are the magnifications of the leftmost panel. Regarding the RD curves of Rippel & Bourdev (2017), we carefully traced the RD curve from the figure of their paper, because the original paper did not provide the exact values at each point. As for the RD curve of Johnston et al. (2017), we used the values provided by the authors via personal communication.

formulated in an equation of the form $\mathrm{bpp} + \lambda \times \mathrm{distortion}$ with a Lagrange multiplier $\lambda$. Needless to say, however, the appropriate value of $\lambda$ depends on the user-chosen weight of the importance for the code-length, which in turn heavily depends on the situation. Training a separate model and conducting a separate hyper-parameter search at every different situation is cumbersome. In order to circumvent this inconvenience, we propose a novel, hyper-parameter free multiplicative loss function that measures the rate-distortion tradeoff.

Equipped with the greater freedom for the choice of PSQE, our ASAP model trained with our multiplicative-loss is not only able to attain compression quality that is on par with the state-of-the-art methods trained with an extensive hyper-parameter search about $\lambda$, but also is able to outperform all the predecessors in terms of the tradeoff ability for almost all region of $\mathrm{bpp}$ constraint. Vice versa, under the same distortion constraint, our codec can compress the image file to as low as $2 \sim 5$ times smaller code in comparison to JPEG. This is $2 \sim 2.5$ times smaller a size in comparison to BPG, which is one of the latest compression algorithms used in industry (see Figs. 2 and 15).

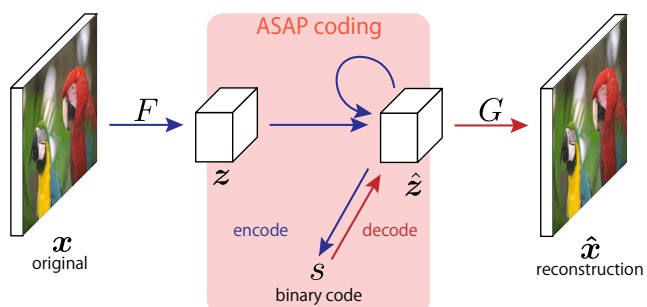

Figure 3: Overall architecture of the proposed model.

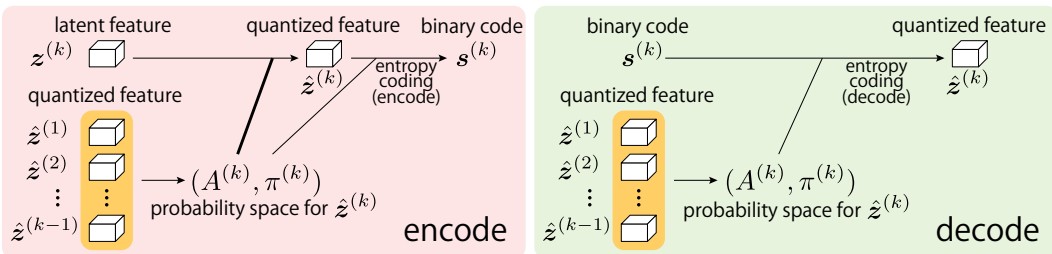

Figure 4: Outline of ASAP coding for image compression.

## 2 METHODS

By all means, we intend our algorithm to be applied to various tasks including audio and video compression. For ease of explanation, however, we will assume for now that our problem is *the compression of image,* and describe the overall flow of our algorithm in the context of image compression. In a nutshell, our method follows the usual routine of transform coding. Throughout, let $\boldsymbol{x} \in \mathbb{R}^{C_0 \times H_0 \times W_0}$ be the original image, where $C_0, H_0, W_0$ respectively represent the number of the channels, the height and the width of the image. The algorithm begins by first applying the transformation function $F$ to $\boldsymbol{x}$ in order to obtain a latent feature representation, $\boldsymbol{z} \in \mathbb{R}^{C \times H \times W}$. The algorithm then quantizes the signal $\boldsymbol{z}$ into a finite space, and then covert the quantized signal $\hat{\boldsymbol{z}}$ to a binary signal $\boldsymbol{s}$ by entropy coding. For the reconstruction, we apply the synthesizer map $G$ to the code $\hat{\boldsymbol{z}}$. That is, $\hat{\boldsymbol{x}} = G(\hat{\boldsymbol{z}})$. Fig. 3 illustrates the overall flow.

In general, we would like to obtain a shorter $s$ with smaller distortion $d(\boldsymbol{x}, \hat{\boldsymbol{x}})$, where $d$ is the measure of the corruption in the reconstruction (e.g., $1 - \text{MS-SSIM}$) (Wang et al., 2004). Our goal is therefore to train the networks that optimize the rate-distortion tradeoff. By construction, the loss of information takes place during the transformation from $\boldsymbol{z}$ to $\hat{\boldsymbol{z}}$. Our ASAP is the invention for this part of the compression. In Sec. 2.1, we describe the distinguishing features of ASAP coding relative to the conventional coding methods. We then describe in Sec. 2.2 the technicalities for the training of the ASAP models in the context of image compression.

### 2.1 THE FEATURES OF ASAP CODING

In this section, we review the basics of entropy coding. Let us assume that $\boldsymbol{z} \in [-1, 1]^{C \times H \times W}$. This $\boldsymbol{z}$ is to be quantized into $\hat{\boldsymbol{z}} \in \mathbb{R}^{C \times H \times W}$ by the discretization of the continuous space $[-1, 1]^{C \times H \times W}$, so that it can be converted into a binary code via invertible compression algorithm.

For the entropy coding of $\hat{\boldsymbol{z}}$, we would have to prepare a probability distribution for $\hat{\boldsymbol{z}}$ that represents its occurring probability. Oftentimes, this occurring probability is modeled by some form of auto regressive model. In our task of compressing the images, we adopted a model used in Nakanishi et al. (2018). We partitioned $\hat{\boldsymbol{z}}$ into $K$ groups $\{\hat{\boldsymbol{z}}^{(k)} \in \mathbb{R}^{C^{(k)} \times H^{(k)} \times W^{(k)}}; k = 1, ..., K\}$ based on a geometric pattern we illustrate in Fig. 5. Let $C^{(k)}, H^{(k)}$, and $W^{(k)}$ be the dimension of the $k$-th group. We then estimated the true occurrence probability as $p^*(\hat{\boldsymbol{z}}) = p^*(\hat{\boldsymbol{z}}^{(1)}) \prod_{k=2}^{K} p^*(\hat{\boldsymbol{z}}^{(k)} | \hat{\boldsymbol{z}}^{(1:k-1)})$

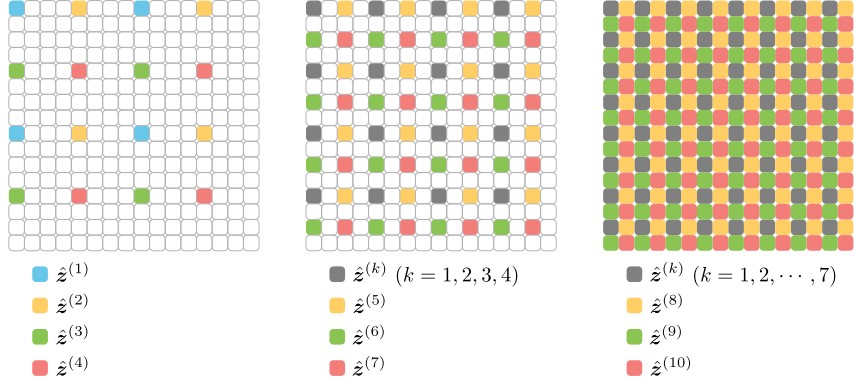

Figure 5: The grouping scheme for the coordinates of $\hat{z}$. This is an imitation of the procedure used in Nakanishi et al. (2018)

by recursively approximating $p^*(\hat{z}^{(k)}|\hat{z}^{(1:k-1)})$ with some probability distribution $\pi(\hat{z}^{(k)}|\hat{z}^{(1:k-1)})$ with discrete support. For ease of notation, we would occasionally use $\pi^{(k)}(\hat{z}^{(k)})$ to denote the same distribution. In this model, we assume that each coordinate of $\hat{z}^{(k)}$ is approximately independent conditional to $\hat{z}^{(1:k-1)}$. This way, we can resort to the power of parallel computation and greatly speed up the algorithm. For more details of this procedure, please see Nakanishi et al. (2018). For the task of compressing other type of data, one can choose different type of partitioning that is natural to the situation.

Our method is novel in that we allow the optimization of the support of $\pi(\hat{z}^{(k)}|\hat{z}^{(1:k-1)})$. We begin by modeling $p^*(\hat{z}_i^{(k)}|\hat{z}^{(1:k-1)})$ with a truncated Gaussian distribution $p(z_i^{(k)}|\hat{z}^{(1:k-1)})$ on $[-1, 1]$ with parameters $\mu_i^{(k)}, \sigma_i^{(k)}$ for each $i$-th coordinate of $z^{(k)}$ using the information of $\hat{z}^{(1:k-1)}$. We then quantize $[-1, 1]$ into the intervals $\{I_i^{(k)}[j]\}_{j=1}^{M_{ki}}$ with the centers $\{A_i^{(k)}[j]\}_{j=1}^{M_{ki}}$ using the quantization width $\{q_i^{(k)}\}_{j=1}^{M_{ki}}$, where $M_{ki}$ is the number of the intervals. The larger the value of $M_{ki}$, finer the quantization. By construction, we construct $\pi(\hat{z}_i^{(k)}|\hat{z}^{(1:k-1)})$ by defining

$$\pi(\hat{z}_i^{(k)}|\hat{z}^{(1:k-1)}) := \sum_{j=1}^{M_{ki}} p(z_i^{(k)} \in I_i^{(k)}[j]|\hat{z}^{(1:k-1)}) \mathbb{1}_{z_i^{(k)}=A_i^{(k)}[j]}. \tag{1}$$

With our method, we can train the objective function about $\mu_i^{(k)}, \sigma_i^{(k)}$, and $q_i^{(k)}$. The ability to optimize $q_i^{(k)}$ is a distinctive feature of our ASAP coding. With this distinctive feature, our ASAP can compress $z$ into a code of much smaller size under the same distortion constraint. By the rate-distortion argument we mentioned in the introduction, this also means that our ASAP can store the data with much smaller distortion under the the same constraint on the code-length.

## 2.2 ASAP CODING FOR IMAGE COMPRESSION

Next, we will explain the technicalities for the optimization of the parameters $\mu_i^{(k)}, \sigma_i^{(k)}$, and $q_i^{(k)}$ in the context of image compression. We will also explain the multiplicative loss, a novel objective function for the training of $R$.

### 2.2.1 ADAPTIVE CONSTRUCTION OF DISCRETIZED SAMPLE SPACE AND PROBABILITY DISTRIBUTION

Fig. 6 illustrates the construction of $\hat{z}^{(k)}$ and $(A^{(k)}, \pi^{(k)})$. From now on, let us write $\pi_i^{(k)}(a) := \pi(\hat{z}_i^{(k)} = a|\hat{z}^{(1:k-1)})$ for $a \in A_i^{(k)}$. In order to recursively construct these probability measures, we trained a set of neural networks that map $\hat{z}^{(1:k-1)}$ to the triplets $\theta_i^{(k)} := (\mu_i^{(k)}, \sigma_i^{(k)}, q_i^{(k)})$. For the base level of $k = 1$, we prepared a trainable triplets $\theta_m^{init} := (\sigma_m^{init}, \mu_m^{init}, q_m^{init})$ for each $m$-th channel

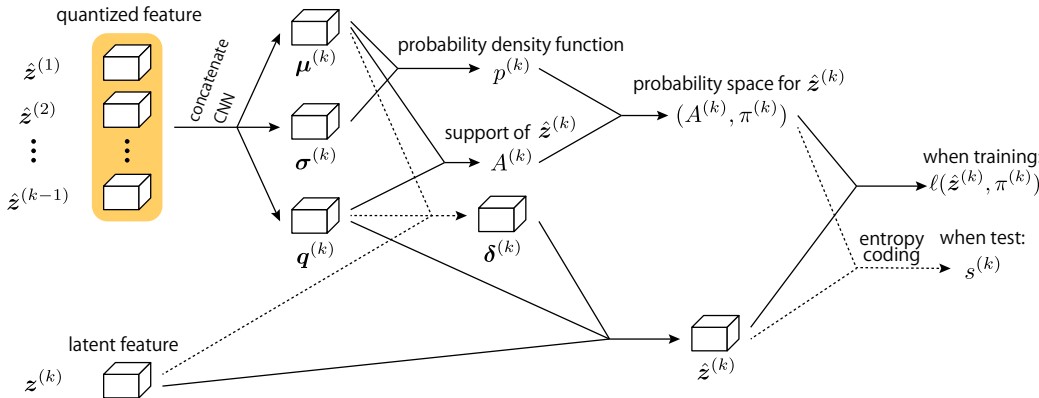

Figure 6: Detailed architecture of the quantizer. The solid line represents the operation subject to back propagation, and the dotted line represents the operation that is not subject to back propagation.

and set $\theta_i^{(1)} = \theta_m^{\text{init}}$ for all $\hat{z}_i^{(1)}$ belonging to the channel $m$ (please see Fig. 14 in the appendix for the detail).

We define $A_i^{(k)}$ as

$$A_i^{(k)} := \left\{ a \Big| a = \mu_i^{(k)} + (u + 0.5)q_i^{(k)}, \ u \in \mathbb{Z}, \ a - 0.5q_i^{(k)} < 1, \ a + 0.5q_i^{(k)} \geq -1 \right\} \quad (2)$$

so that $I_i^{(k)}[j] = \left[ A_i^{(k)}[j] - 0.5q_i^{(k)}, A_i^{(k)}[j] + 0.5q_i^{(k)} \right)$. This indeed amounts to discretizing the interval $[-1, 1]$ using the subintervals of width $q_i^{(k)}$; $M_{ki}$ is determined naturally from this constraint. To each subinterval, we would assign the probability mass computed with the formula

$$\pi_i^{(k)}(a) := \int_{a-0.5q_i^{(k)}}^{a+0.5q_i^{(k)}} p_i^{(k)}(x)\mathrm{d}x, \quad (3)$$

where $p_i^{(k)}$ is a truncated normal distribution with the parameters $(\mu_i^{(k)}, \sigma_i^{(k)})$ that is defined over the domain $\left[ \underline{a}_i^{(k)} - 0.5q_i^{(k)}, \overline{a}_i^{(k)} + 0.5q_i^{(k)} \right)$. We are aware that this is not technically the faithful discretization of the interval $[-1, 1]$; this compromise is for ease of implementation. If $I_i^{(k)}$ perfectly partitions $[-1, 1]$, the support of $p_i^{(k)}$ will be exactly $[-1, 1]$ [1]. By construction, our $\pi_i^{(k)}$ satisfies the required properties of the probability mass function. By appealing to the basic result of information theory, any signal sampled from the distribution $\pi_i^{(k)}$ can theoretically be encoded in a code-length:

$$\ell(\hat{\boldsymbol{z}}^{(k)}; \pi^{(k)}) := - \sum_{i=1}^{C^{(k)} H^{(k)} W^{(k)}} \log_2 \pi^{(k)}(\hat{z}_i^{(k)}|\hat{\boldsymbol{z}}^{(1:k-1)}). \quad (4)$$

In order to map $z_i^{(k)}$ to $A_i^{(k)}$, we simply pick the member of $A_i^{(k)}$ that is closest to $z_i^{(k)}$; this member would be the value we assign to $\hat{z}_i^{(k)}$. By construction, the quantization error $\hat{z}_i^{(k)} - z_i^{(k)}$ is bounded from above by $q_i^{(k)}/2$.

**Training of the neural models**

In order to make the training of the neural networks possible, we would take advantage of the fact that we can always write

$$\hat{z}_i^{(k)} = z_i^{(k)} - \delta_i^{(k)}q_i^{(k)}. \quad (5)$$

for some $\delta_i^{(k)} \in [-0.5, 0.5]$. We can therefore interpret $\delta_i^{(k)}$ as a noise and regard the equation above as a probabilistic model for $\hat{z}_i^{(k)}$ with a trainable noise parameter $q_i^{(k)}$. Indeed, the demand

---

[1]In fact, in order to avoid the creation of the interval with too small a mass, we have occasionally augmented $p_i^{(k)}$ and renormalized the integrals.

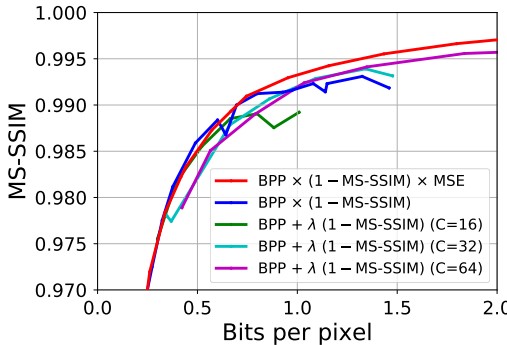

Figure 7: Rate-distortion tradeoff curve for the ASAP model trained with different objective functions. For the Multiplicative loss, Multiplicative rate-distortion trade-off function. For the additive costs, we run experiments with few selected values of $C$ and varied the choice of $\lambda$ only. This is because the computational cost would explode if we are to do the parameter search of $\lambda$ for large number of $C$. We should add, as a cautionary remark, that the results shown hereof are the results obtained with less number of iterations (0.1M) than the results in Figure 2 (0.3M).

for smaller distortion (loss of information) would prefer smaller $q_i^{(k)}$ and hence smaller variance for noise. On the contrary, the demand for shorter code-length would prefer coarser discretization of the sample space and hence larger $q_i^{(k)}$. This is indeed the very manifestation of the rate-distortion tradeoff for the training of $q_i^{(k)}$.

### 2.2.2 MULTIPLICATIVE RATE-DISTORTION TRADEOFF FUNCTION

In this section, we will finally describe the objective function by which we would train the ASAP model we introduced above. In order to make comparisons, we would introduce our objective function along with its variation and the classic counterpart:

$$\text{bpp} + \lambda \times (1 - \text{MS-SSIM}) \quad (\lambda \geq 0) \tag{6}$$
$$\text{bpp} \times (1 - \text{MS-SSIM}) \tag{7}$$
$$\text{bpp} \times (1 - \text{MS-SSIM}) \times \text{MSE} \quad \text{(proposed method)}, \tag{8}$$

where $\text{bpp} := \frac{1}{H_0 W_0} \sum_{k=1}^{K} \ell(\hat{\boldsymbol{z}}^{(k)}; \pi^{(k)})$. An apparent disadvantage of Eq. (6) is that one must search for different appropriate $\lambda$ for different constraint on the compression rate (code-length).

In order to do away with the computationally heavy parameter search, we took advantage of our observation that the relation $\text{bpp} \times (1 - \text{MS-SSIM}) = \text{const}$ approximately holds in experiments, and first adopted this function as a candidate for the objective function (Eq. (7)). As we show in Fig. 7, our ASAP trained with this objective function performs almost as well as the ASAP trained with the Eq. (6) with the optimal $\lambda$ for all choice of the compression rate. However, as we can see in Fig. 7, the MS-SSIM value for both Eqs. (6) and (7) plateaus at some value less than 1. We resolved this issue by modifying Eq. (7) to Eq. (8). This solution works because the inclusion of the MSE cost would create still another pressure to the training process. By this modification, we were able to produce good results in the region of large bpp as well.

## 3 RELATED WORKS

Application of the neural networks to the compression of image dataset is popular these days. Most studies choose (recurrent) autoencoder based on CNN as the main architecture. By its construction, the coordinates of the output from such autoencoder will have strong spatial correlation. This tendency should be particularly true for the network trained for the compression of the images. Moreover, the local correlative relations among the coordinates can greatly vary across the locations and the channels, not to mention across the types of images. In order to optimize the rate-distortion trade off, one must therefore strive to choose appropriate quantization width for different situations.

Indeed, this effort must be jointly planned and implemented with the optimization of the parameters of the autoencoder, because the the correlations amongst the coordinates of the feature vector is obviously dependent on the behavior of the autoencoder. The work of Ballé et al. (2017) is one of the pioneer studies that trained PSQE together with the autoencoder. Ballé et al. (2018) further improved their work by introducing an additional latent variable that governs the spatial relation among the quantization probability distributions of the latent spaces.

The optimization of the quantization process is a challenging problem in general, and numerous studies proposed different approaches to this problem. Toderici et al. (2015; 2017); Johnston et al. (2017); Ballé et al. (2017; 2018) tackled this problem by interpreting the quantization process probabilistically. In these approaches, the use of probabilistic noise can de-stabilize the training process; this tends to be particularly true for variational inferences. The works of Theis et al. (2017); Nakanishi et al. (2018); Mentzer et al. (2018) fall in to this category. Mentzer et al. (2018) in particular deploys a set of trainable parameters $C = \{c_1, \cdots, c_L\}$ from which to sample $\hat{z}$ with the encoding rule $\hat{z}_i = \arg\min_j \|z_i - c_j\|$. The compression method of Mentzer et al. (2018), however, trains $C$ of fixed length (i.e., the number of bins). On the other hand, in our study, the number of bins is a function of quantization width. Our method therefore optimizes the number of bins together with the quantization width.

Numerous types of objective functions are also used for the training of the network for compression task. Most of the previous works like Ballé et al. (2017); Rippel & Bourdev (2017); Mentzer et al. (2018) use the objective function of type (6). There is also a variety of distortion measure, such as MS-SSIM, $L_2$ loss, $L_1$ loss, as well as their weighted mixtures. Some studies also use GANs' loss (Rippel & Bourdev, 2017; Agustsson et al., 2018). Johnston et al. (2017) uses a multiplicative loss of the form SSIM $\times L_1$. As mentioned in the previous sections, our multiplicative loss is based on the observation that the empirical trade-off functions resemble a function of the form bpp $\times (1 - \text{MS-SSIM}) = \text{const}$. It shall be emphasized that we were able to produce superior result using a single, hyper-parameter free objective function.

## 4 EXPERIMENT

We evaluated the performance of our method on the benchmark image datasets. We trained our proposed model over 0.3M iterations with batchsize 25. For the optimization, we used Adam (Kingma & Ba, 2015) with AMSGrad (Reddi et al., 2018). As for the hyper-parameters of Adam, we set $\alpha = 0.001, \beta_1 = 0.9, \beta_2 = 0.999$, and linearly decayed $\alpha$ from the 0.225M-th iteration so that it will reach 0 at the end of the training. For the update of the parameters, we applied $L_2$ regularization with weight decay of rate 0.00001, and clipped the magnitude of each coordinate of the gradient at the value of 5. As a similarity measure between two images, we used MS-SSIM (Wang et al., 2004), which is considered by the community to have close correlation with the subjective evaluation of humans. Because MS-SSIM is originally intended for the evaluation of gray scale measures, we computed the MS-SSIM separately for each of RGB channels and reported their average. For the training, we used ILSVRC2012 dataset (ImageNet) (Russakovsky et al., 2015). In order to produce the rate distortion (RD) curve, we trained a model for each choice of $C$, the number of channels for the latent feature. It shall be reminded that $C$ affects the code-length via entropy coding. For the performance evaluation, we used Kodak[2] dataset and RAISE-1k dataset (Dang-Nguyen et al., 2015).

### 4.1 PERFORMANCE

Kodak dataset is a widely used dataset for the performance evaluation of compression methods. Fig. 2 compares the RD curves for different compression methods. In terms of MS-SSIM, our method was slightly better than the method trained with the *best* fixed quantization size, which was found through an extensive grid search. Again in terms of MS-SSIM, our method outperformed the model of Ballé et al. (2018) trained to optimize MS-SSIM ("Balle for MS-SSIM"), as well as the same model trained to optimize mean squared error (MSE) ("Balle for MSE"). In terms of PSNR, our method performed better than "Balle for MS-SSIM", not to mention JPEG. However, it was not able to perfrom better than "Balle for MSE" in terms of PSNR.

---

[2]http://r0k.us/graphics/kodak/.

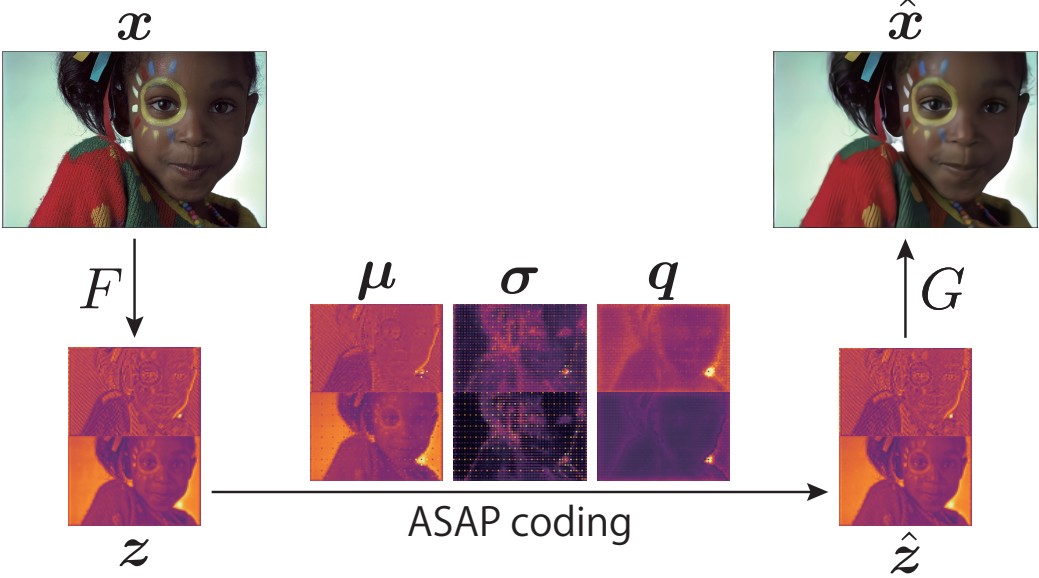

Figure 8: The heat map of $\mu$, $\sigma$, $q$ produced by a trained ASAP model for a Kodak image. The stacked pair of images corresponds to the heat maps of a selected pair of channels. Notice that the network is choosing larger value of $q$ for a channel containing more information.

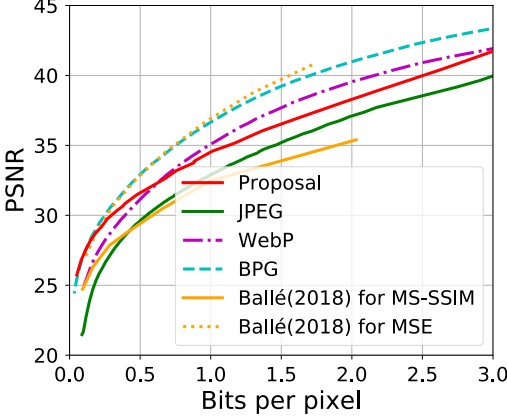

Figure 9: Performances of various compression methods, evaluated in terms of PSNR.

For the visuals of the reconstructed images, please see Figs. 1 and 16–18. Figs. 19–24 illustrate regions of medium-high bpp. Our method outperforms BPG on the medium-high bpp region.

Fig. 8 illustrates the heat map of $(\mu_i^{(k)}, \sigma_i^{(k)}, q_i^{(k)})$ produced for a Kodak image by a trained ASAP model. The signals from each $z^{(k)}$ was re-aligned in the way that respects the original geometrical relations prescribed by the pattern in Fig. 5. A pair of stacked images is a pair of images from two different channels. Fig. 8 features several interesting properties of our ASAP coding. Firstly, as we can readily see in the figure, the 'bottom' channel is capturing more contextual information than the 'top' channel. Note that our ASAP is assigning larger values of $q$ in general to the bottom channel than to the top channel. We can also see that the algorithm is assigning larger value of $q$ to the region within the picture with relatively small information. Lastly, the value of $\sigma$ assigned by the algorithm to each coordinate is reflective of the algorithm's *confidence* for the compression of the information on the coordinate. By construction, $z^{(1)}$ is a group for which the algorithm would have least confidence because it is the base level of the recursive inference. The bright points (representing large value) scattered across the heat map in grid-pattern correspond to $z^{(1)}$.

Kodak dataset consists of only 24 images. To rule out the possibility that our model is overfitted to the Kodak dataset by chance, we also evaluated the performance of our models using RAISE-1k dataset that consists of 1000 raw images (Fig. 15). We can confirm the superiority of our method over all the current contemporaries on RAISE-1k dataset as well. It shall be noted that our method is outperforming other methods not only in the low bit rate region, but also in the high bit rate region as well.

## 4.2 THE EFFECTIVENESS OF THE MULTIPLICATIVE LOSS

We conducted still another comparative experiment in order to evaluate the effectiveness of using the multiplicative loss. Fig. 7 shows the RD curves of our model trained with the three different objective functions we introduced in Sec. 2.2.2 (Eqs. (6)–(8)). We used the same experimental setup as the one we used in Sec. 4.2 except that we stopped the iteration at 0.1M-th step and initiated the linear decay of $\alpha$ from 0.075M-th step. To be more fair, we tested the performance of Eq. (6) using multiple good values of $\lambda$. We can confirm in Fig. 7 that the multiplicative loss outperforms all other loss functions.

We shall also note that the model trained to optimize our multiplicative loss yielded the results with higher MS-SSIM than the model trained to directly optimize the MS-SSIM loss. We shall emphasize that our multiplicative loss does not require the hyperparameter search about $\lambda$ for each bit rate.

## 4.3 ABLATION STUDY WITH SMALL ASAP MODEL

We conducted an ablation study to assess the effect of making the quantization width adaptive for each latent feature. We conducted the compression with a fixed quantization width for all latent features and compared the result with our ASAP coding. For the size of the the common quantization width, we conducted a grid search and selected the width that achieved the best MS-SSIM score. Figure 10a compares the result of our algorithm against those of different choices of *fixed* quantization width. We can see in the plot that our algorithm performs equally well with the result with the optimal *fixed* quantization width. Co-incidentally, there was a choice of *fixed* quantization width that works equally well with the adaptive quantization width. We also conducted a set of experiments with a smaller version of our model, in which we reduced the number of channels for each convolution layer and replaced each residual block with a single convolution layer. The figure 10b shows the results of the experiments with the *small* model. As we can see in the figure, the adaptive quantization width performs better than all choices of *fixed* quantization width we tested. All in all, we were able to confirm through the ablation study that our adaptive quantization width performs equally well or better in comparison to the optimal *fixed* width quantization that was found through a grid-search.

## 5 CONCLUSION

In this study, we proposed ASAP, a novel lossy compressor based on a neural architecture that allows the training of quantization width. We also proposed multiplicative loss, a novel hyper-

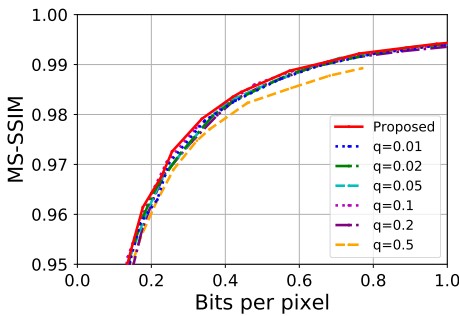 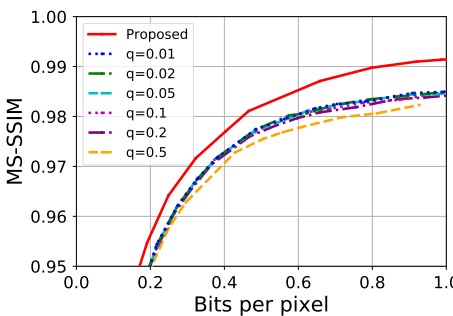

(a) The effect of the adaptive quantization width for the model used for the benchmark study

(b) The effect of the adaptive quantization width for a small model

Figure 10: (a) Ablation study for the effect of adaptive quantization width. The adaptive quantization width works equally well with the *best* fixed quantization width, which was found through an extensive grid search. (b) The same ablation study conducted with a smaller version of the ASAP model (normal convolution layers in place of resblocks, smaller number of channels). The adaptive quantization width works better than the *best* fixed quantization width.

parameter free objective function that can be used to train the compressor with better rate-distortion tradeoff. Our study has shown that there is still a room for improvement in the neural network based approaches to compression, which once seemed to have reached its limit. The application of our ASAP is not be limited to image compression. The framework of ASAP and the idea of the multiplicative loss shall be applicable to various industrial needs in which the rate distortion tradeoff is an important factor.

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

# A    THE NETWORK ARCHITECTURE OF ASAP CODING

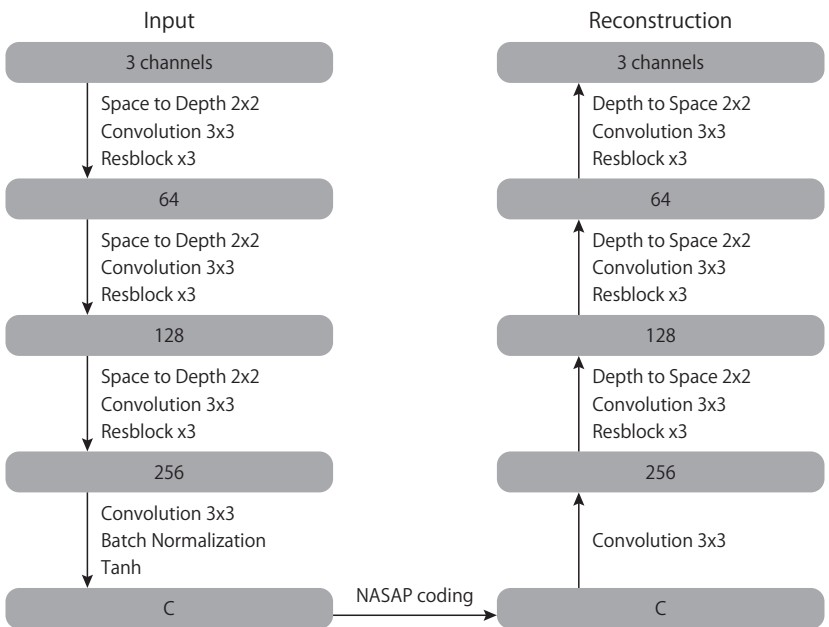

Figure 11: Detail architecture of proposed model.

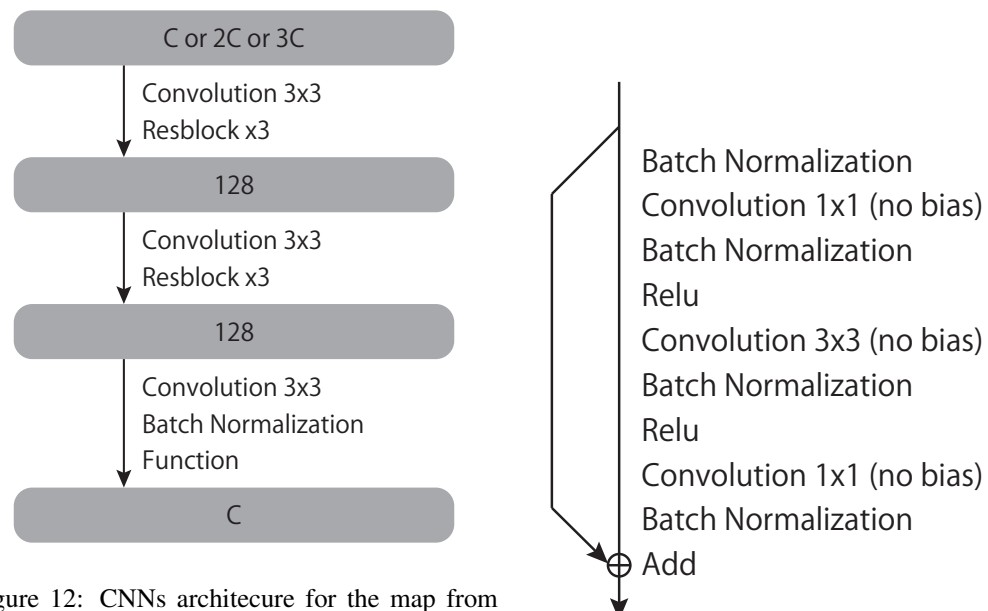

Figure 12: CNNs architecure for the map from $\hat{z}^{(1:k-1)}$ to $\mu^{(k)}, \sigma^{(k)}$, and $q^{(k)}$. The map designated as "Function" stands for an element-wise non-linear function, which in our case is tanh for $\mu$ and $2 \times \mathrm{sigmoid} + \epsilon$ for $\sigma$ and $q$, where $\epsilon$ is a small positive constant.

Figure 13: Detail architecture of one ResBlock.

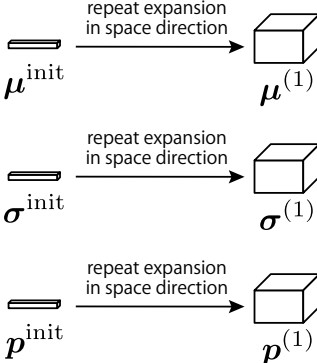

Figure 14: Construction of $(\boldsymbol{\mu}^{(1)}, \boldsymbol{\sigma}^{(1)}, \boldsymbol{q}^{(1)})$. One set of parameters was created for each channel, and same parameter set was assigned to all the coordinates belonging to same channel.

# B  APPENDIX RESULTS

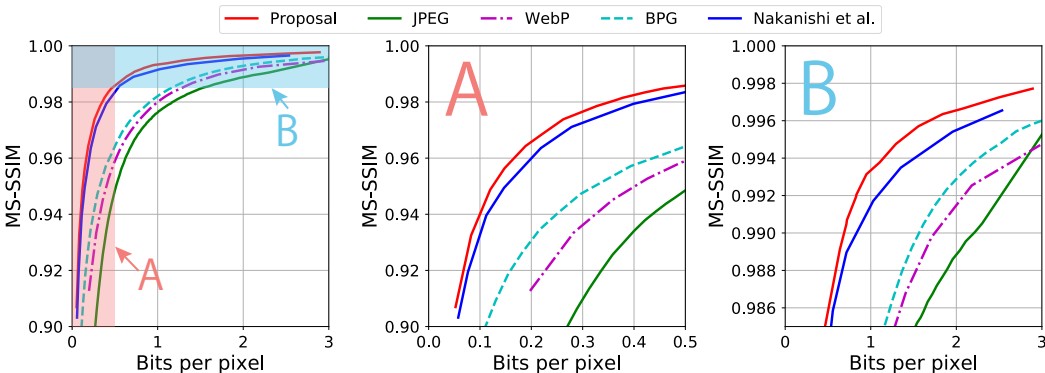

Figure 15: Rate-distortion tradeoff curves evaluated for different methods on resized RAISE-1K (Dang-Nguyen et al., 2015) dataset.

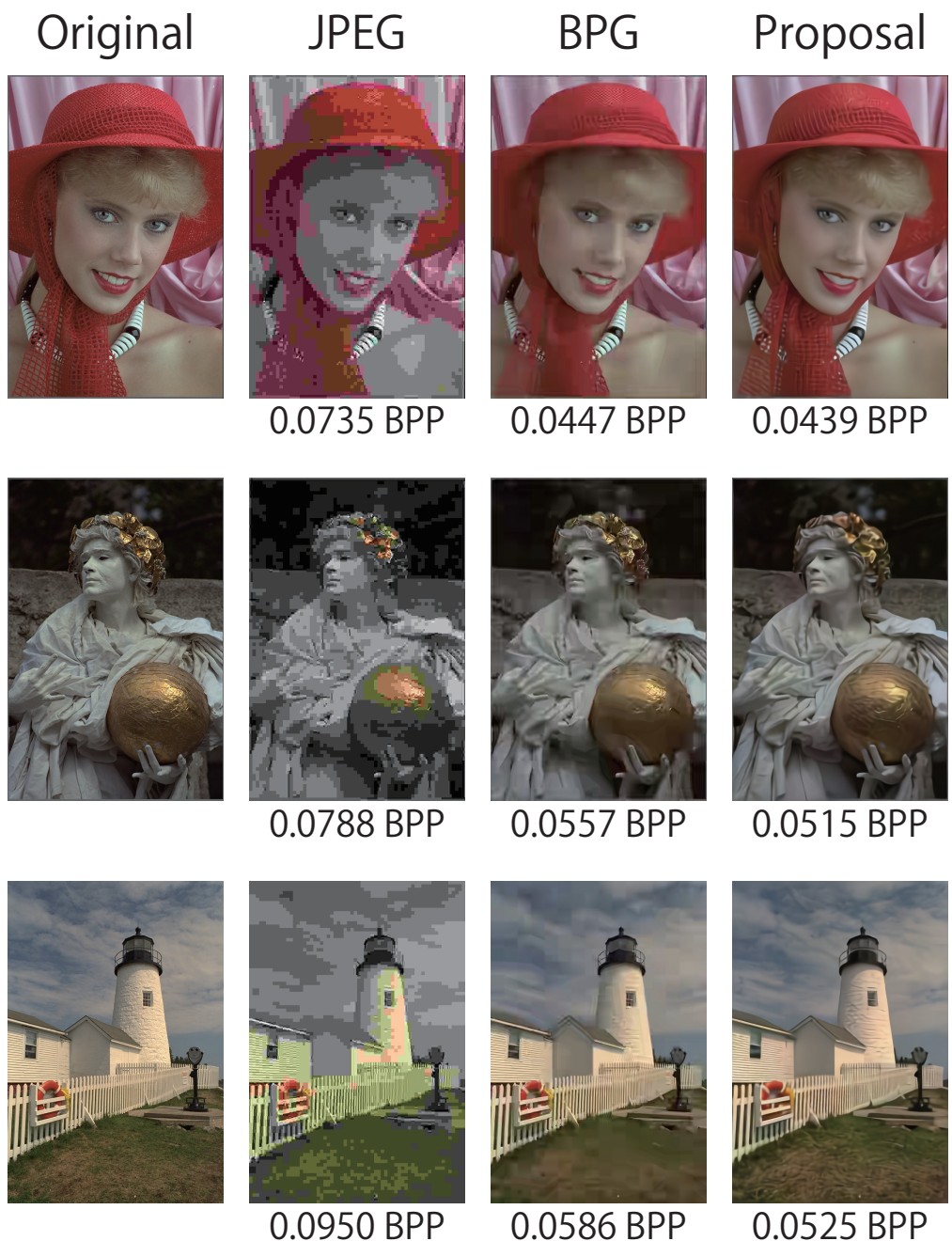

Figure 16: Images reconstructed with various coding methods.

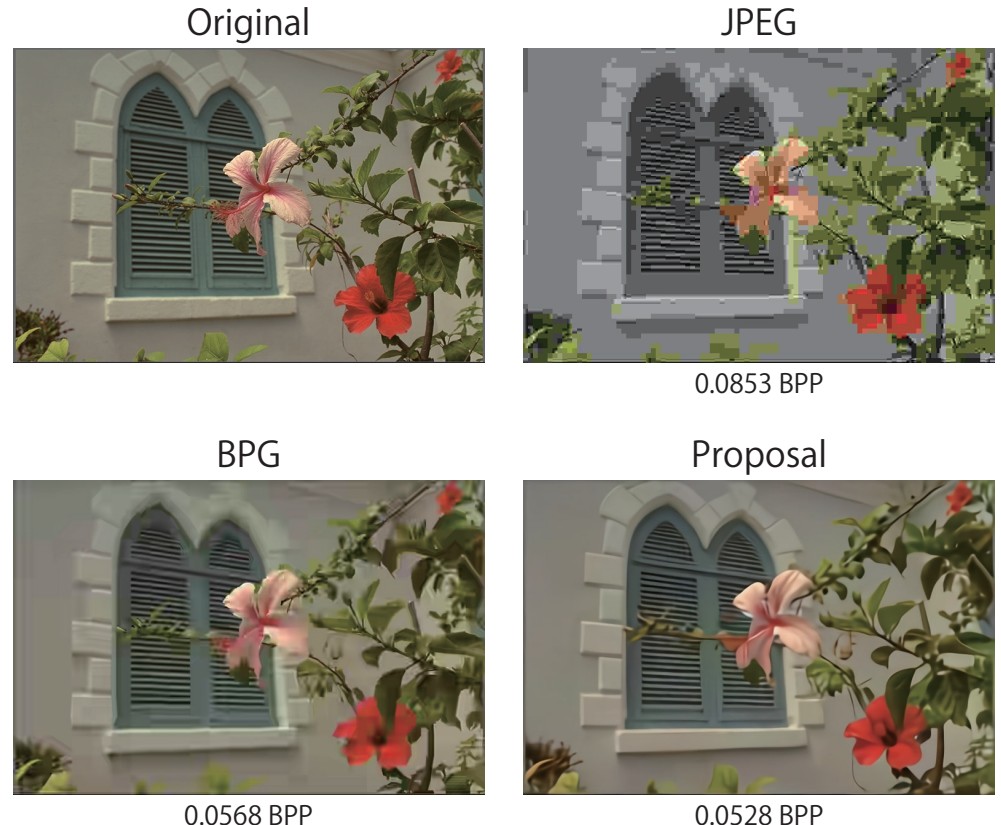

Figure 17: Images reconstructed with various coding methods (con'd).

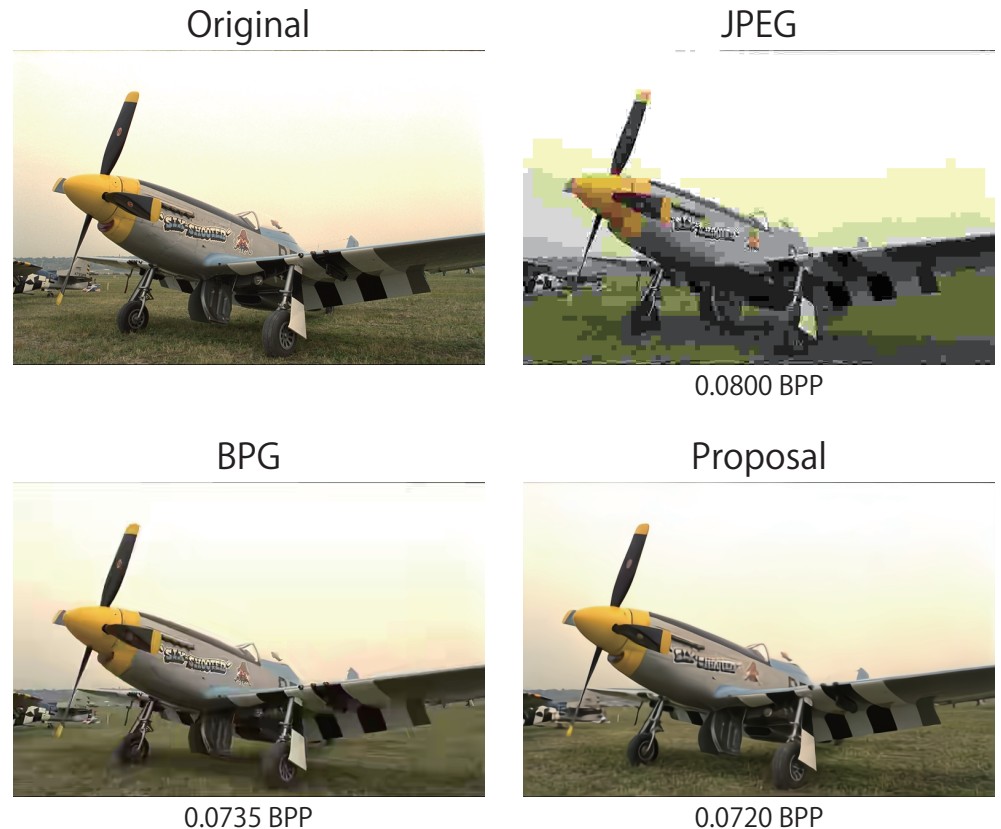

Figure 18: Images reconstructed with various coding methods (con'd).

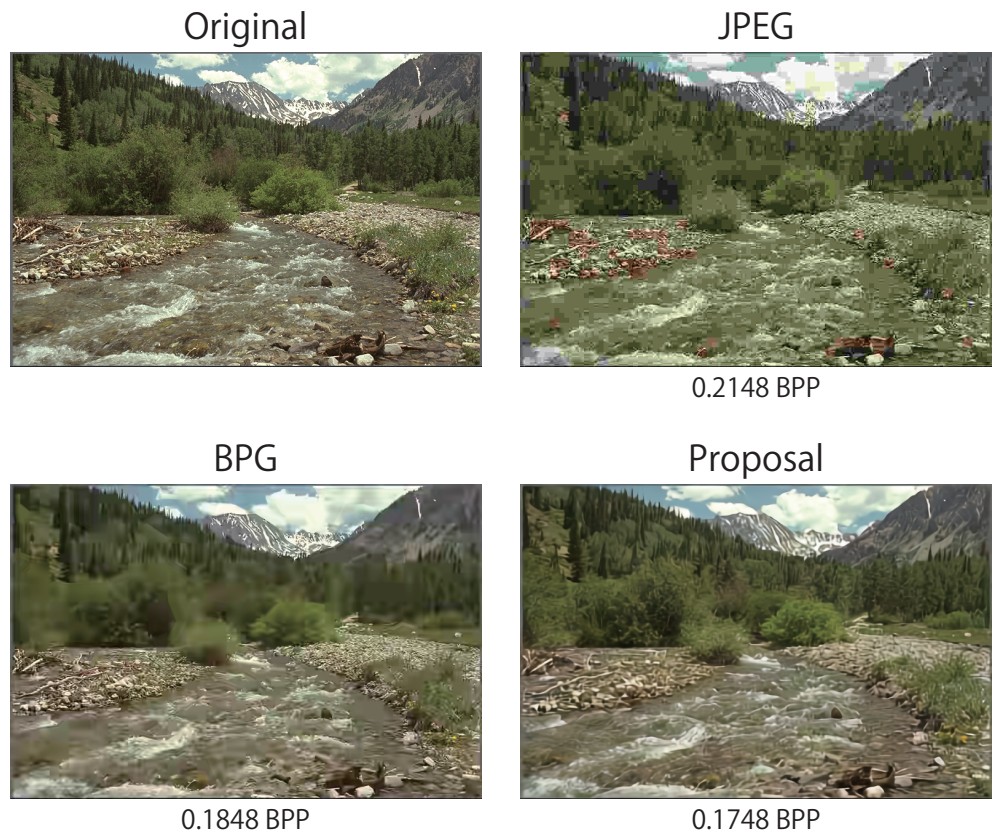

Figure 19: Comparison of the visual results of the reconstruction for middle-high bpp range.

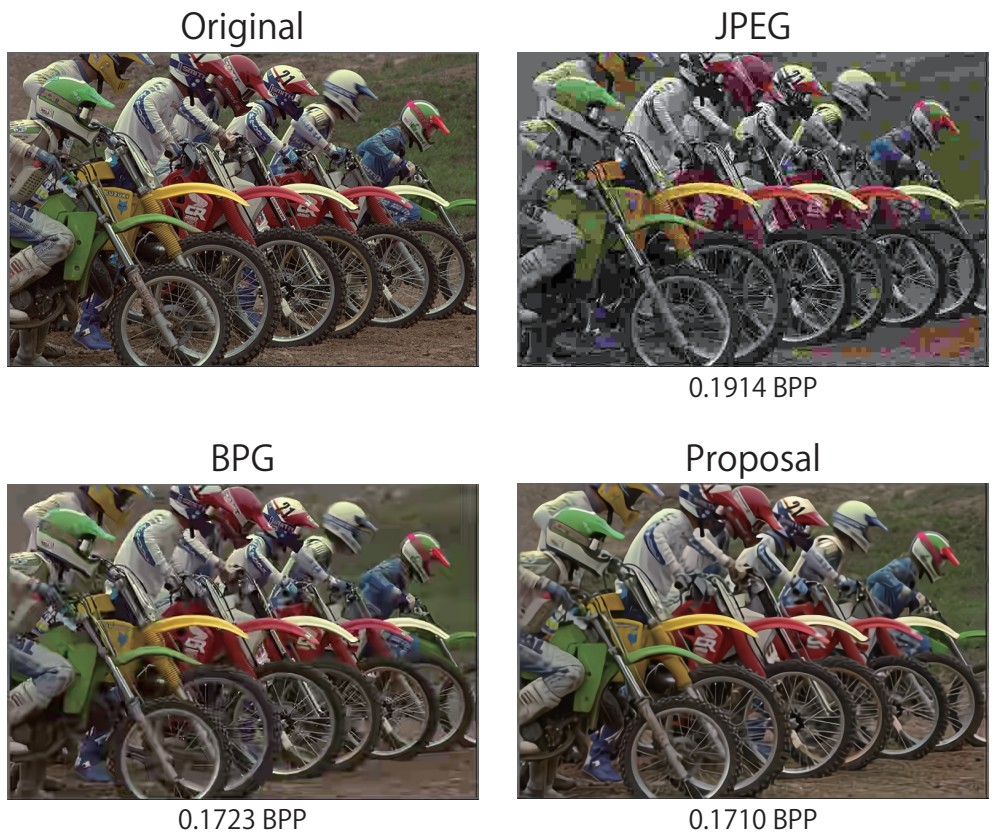

Figure 20: Comparison of the visual results of the reconstruction for middle-high bpp range (con'd).

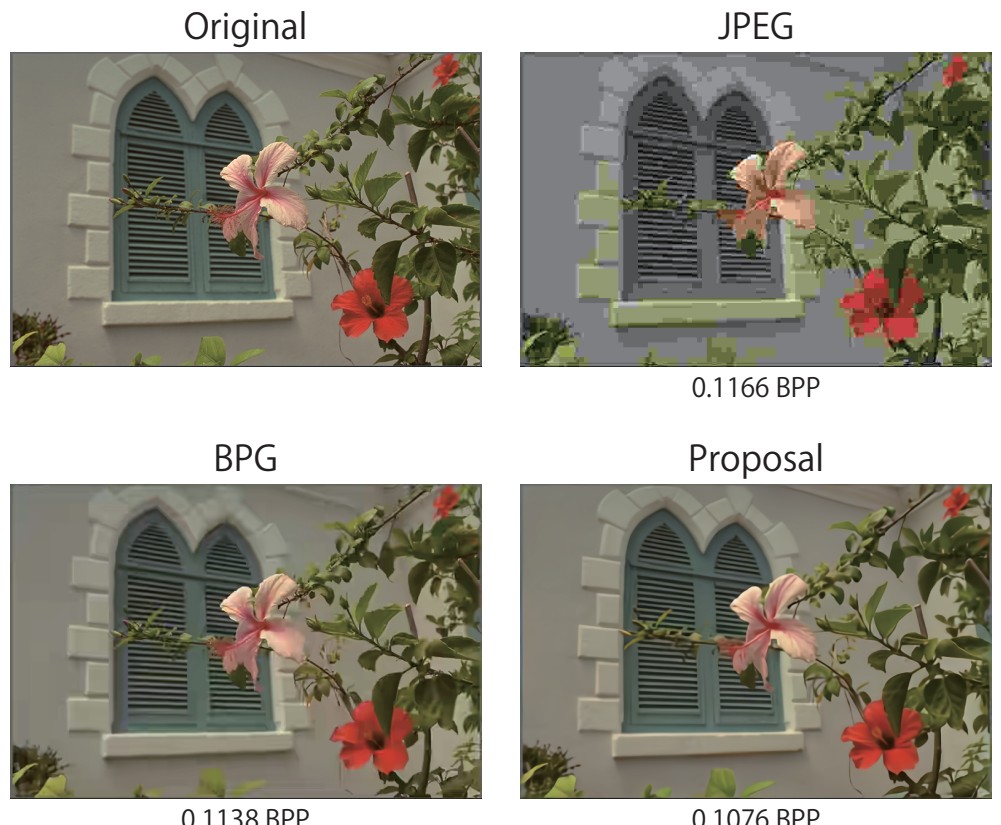

Figure 21: Comparison of the visual results of the reconstruction for middle-high bpp range (con'd).

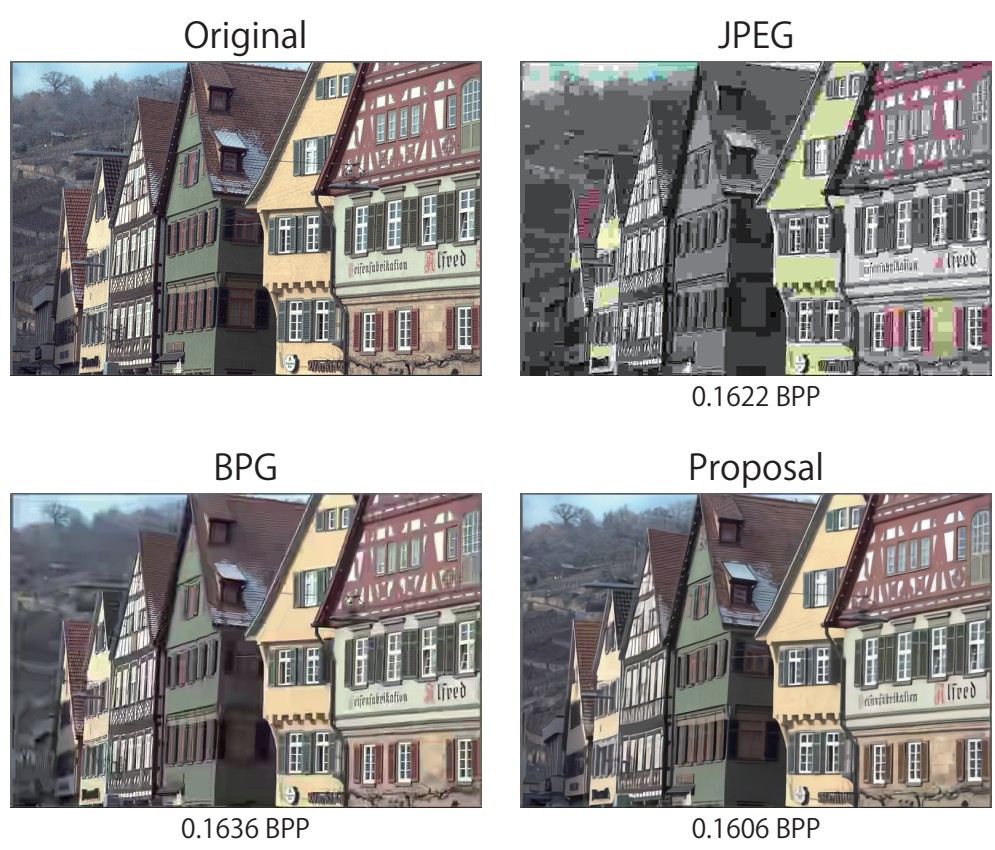

Figure 22: Comparison of the visual results of the reconstruction for middle-high bpp range (con'd).

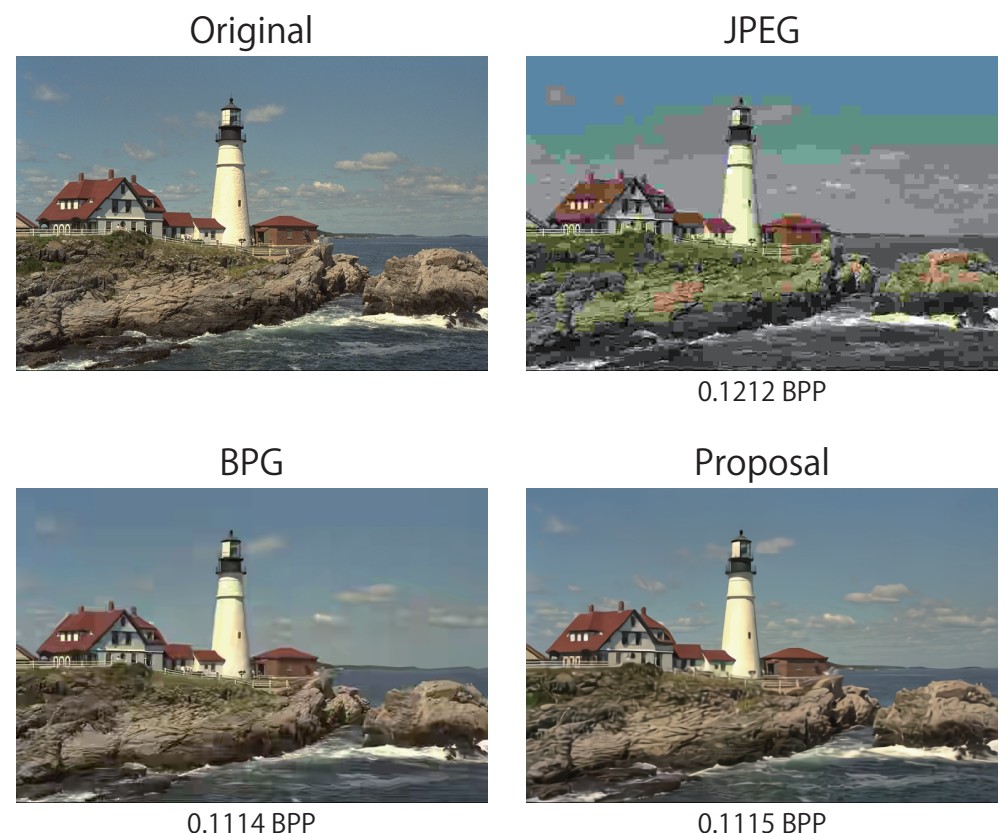

Figure 23: Comparison of the visual results of the reconstruction for middle-high bpp range (con'd).

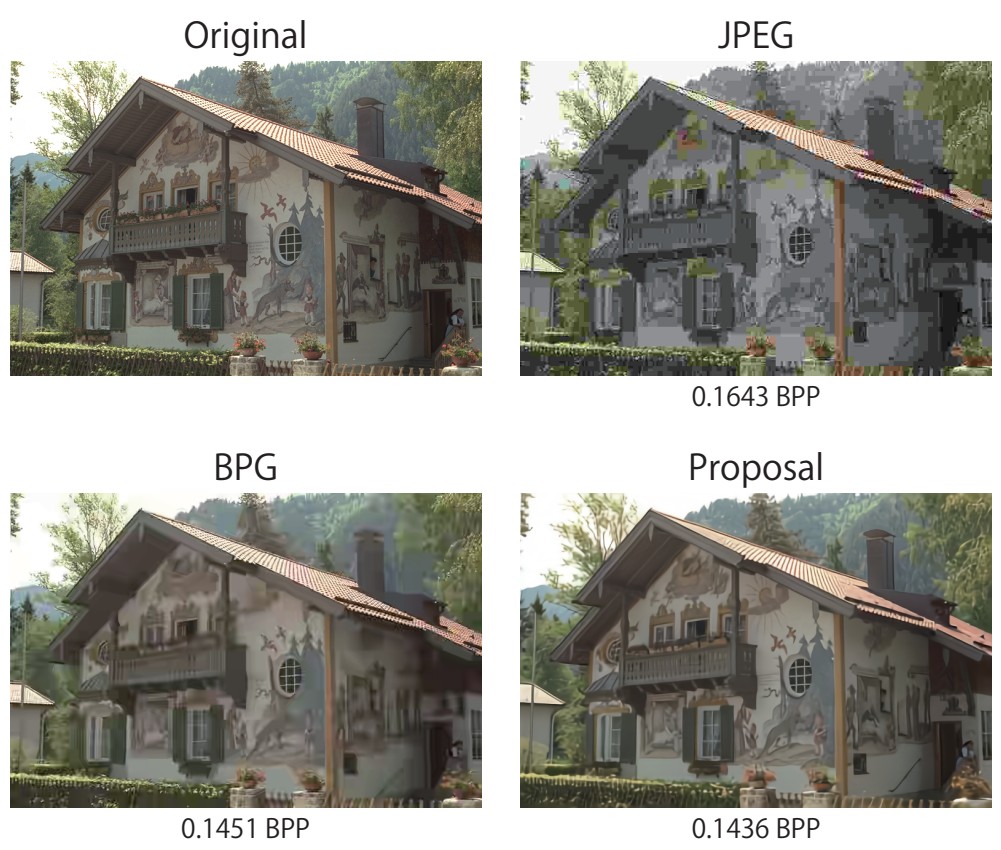

Figure 24: Comparison of the visual results of the reconstruction for middle-high bpp range (con'd).

