# OpenReview forum: "Adaptive Sample-space & Adaptive Probability coding: a neural-network based approach for compression"
_ICLR.cc/2019/Conference_

### Official Review · AnonReviewer2 · 2018-11-01
**Is it the loss or the quantization that matters?**

**Rating:** 5
**Confidence:** 4

**Review:**

The paper proposes Adaptive Sample-space & Adaptive Probability (ASAP) coding for image compression based on neural networks. In contrast to most prior methods, which adhere to a fixed quantization scheme (i.e. with fixed number of quantization levels, and fixing the level themselves), the proposed method jointly learns a probability model of the quantized representation (the bottleneck of an autoencoder model) for coding and a corresponding adaptive quantization scheme. The distribution of each entry in the bottleneck before quantization is modeled as a Gaussian, whose mean and variance are predicted by a neural network conditionally on bottleneck entries on a grid at different scales (similar as in Nakanishi et al. 2018). The same network also predicts quantization intervals to adaptively quantize the respective entry of the bottleneck. Together, the predicted means, variances, and quantization intervals are used to obtain an estimate of the code length. The proposed compression networks are trained with a novel multiplicative loss, showing clear improvements over prior methods Rippel & Bourdev 2017, Nakanishi et al. 2018 on the Kodak and Raise1k data sets in terms of MS-SSIM.

Pros:

The results presented in this paper seem to be state-of-the-art, and innovation on quantization, which has not attracted a lot of attention in the context of neural network-based image compression is a welcome contribution. The method also seems to outperform the recent method [1], which should be included for comparison.

Questions:

A major issue is, however, that it is unclear from the results whether the gains are due to the novel quantization system, or due to the novel loss. From Fig. 7 it looks like the loss BPP + \lambda (1-MS-SSIM) (assuming the formula in (6)) is correct, and the legend in Fig. 7 incorrect) that is used in most other works performs essentially on par with  Rippel & Bourdev 2017, Nakanishi et al. 2018. For example at 1 bpp, this loss yields an MS-SSIM of 0.992 which is essentially the same as Rippel & Bourdev 2017, Nakanishi et al. 2018 obtain, cf. Fig. 2. To show that the improvement is due to the learned quantization and not just because of the loss (8) an ablation experiment should be done. One could e.g. train the proposed method with the same predictor, but without the employing the learned quantization scheme, and compare to the results obtained for the proposed method.

Furthermore, a better understanding of the loss (8) would be desirable. How is the MSE factor justified?

Also, it would be good to present visual examples at rates 0.1-1.0 bpp. All visual examples are at rates below 0.08 bpp, and the proposed method is shown to also outperform other methods at much higher rates.


[1] Ballé, J., Minnen, D., Singh, S., Hwang, S.J. and Johnston, N. Variational image compression with a scale hyperprior. ICLR 2018.

---

> ### Author Response · Authors · 2018-11-26
> **Thank you very much for comments and suggestions!**
>
> Thank you very much for comments and suggestions.  We made revisions to reflect the suggestions made and to resolve the concerns raised.  We would also like to provide responses to the comments below:
>
> "A major issue is, however, that it is unclear from the results whether the gains are due to the novel quantization system, or due to the novel loss. From Fig. 7 it looks like the loss BPP + \lambda (1-MS-SSIM) (assuming the formula in (6)) is correct, and the legend in Fig. 7 incorrect) that is used in most other works performs essentially on par with  Rippel & Bourdev 2017, Nakanishi et al. 2018. For example at 1 bpp, this loss yields an MS-SSIM of 0.992 which is essentially the same as Rippel & Bourdev 2017, Nakanishi et al. 2018 obtain, cf. Fig. 2. To show that the improvement is due to the learned quantization and not just >because of the loss (8) an ablation experiment should be done. One could e.g. train the proposed method with the same predictor, but without the employing the learned quantization scheme, and compare to the results obtained for the proposed method."
>
> This was in fact a concern for the other reviewers too, and we conducted an ablation study to assess the effect of ‘dropping’ the adaptive quantization width.  That is, we conducted a set of experiments in which we used a fixed quantization width for all latent features.   As we can see in the newly added figure,  the algorithm with adaptive quantization width was able to perform equally well with the ‘best’ fixed-width quantization in terms of MS-SSIM.   We also conducted a same set of ablation studies with a smaller model (no resblock, smaller # channels). For this ablation study,  adaptive quantization width worked much better than the fixed quantization.
>
>
> "Furthermore, a better understanding of the loss (8) would be desirable. How is the MSE factor justified?"
>
> Unfortunately, we are unable to provide a solid answer to this question yet.
> Although not too intuitive, the training with the MSE-included multiplicative loss was in fact smoother than the training with MS-SSIM loss. Empirically,  the presence of MSE often seemed to help the training process evade the local minima in the MS-SSIM landscape. As we can see in the figure added in the Appendix, the rate-distortion curve is much smoother for the compression results produced by the model trained with the MSE-included multiplicative loss. However, when we checked the effect of the multiplicative loss on the difference between the test result and the training result,  it seemed that we can at least say that the inclusion of MSE does not have a `regularization effect`.
>
>
> "Also, it would be good to present visual examples at rates 0.1-1.0 bpp. All visual examples are at rates below 0.08 bpp, and the proposed method is shown to also outperform other methods at much higher rates."
>
> We added a set of visual results for the medium-high bpp (0.1~)  compressions in the revision.

---

> > ### Comment · AnonReviewer2 · 2018-11-29
> > **Response to rebuttal**
> >
> > Thanks for the detailed response. It is interesting to see that the proposed quantization helps on the small network. However, I still think there are some aspects that should be explored more like the role of the loss terms.

---

> > > ### Author Response · Authors · 2018-12-04
> > > **Thank you!**
> > >
> > > In addition to the added analysis and observations we stated in the revision, we are also inferring from our results that the energy landscape of MS-SSIM contains multiple local extrama and it is, at least for the dataset we have studied, difficult to optimize.  In fact, the model optimized for MS-SSIM is worse in terms of MS-SSIM than the model optimized for proposed multiplicative loss not only on test set, but notably also on training set.
> > >
> > > proposed multiplicative loss (bpp x (1 - msssim) x mse):
> > > train/bpp   train/msssim    train/mse   val/bpp     val/msssim  val/mse
> > > 0.939190    0.006595        57.649879   0.953790    0.007047    64.067696
> > >
> > > bpp x (1 - msssim):
> > > train/bpp   train/msssim    train/mse   val/bpp     val/msssim  val/mse
> > > 0.913846    0.008405        106.999641  0.926681    0.008613    99.586456
> > > 0.955831    0.008065        96.597839   0.974577    0.008393    100.409378
> > >
> > > bpp + lmb x (1 - msssim):  ※ C=32
> > > train/bpp   train/msssim    train/mse   val/bpp     val/msssim  val/mse
> > > 1.077895    0.006823        85.435677   1.091407    0.007135    88.454208
> > > 0.859573    0.008996        109.393616  0.860719    0.009333    108.734512
> > > (evaluation for a training sample and a validation sample of ImageNet )
> > >
> > > The evaluations on validation set are identical to the ones shown in Fig.7.
> > > Note that, MS-SSIM score on the validation set is not only smaller in general for the model optimized for proposed multiplicative loss,
> > > the difference between the training and validation is not also larger for the model optimized for proposed multiplicative loss than the model optimized for MS-SSIM.  This suggests that the model is not doing well in the direct optimization about MS-SSIM score on the training set; that is, MS-SSIM is on its own a difficult cost function to train with.
> > >
> > > Our observations and claims are also supported by the fact that the MS-SSIM does not increase smoothly with bpp (Fig 7).  As  discussed in https://arxiv.org/pdf/1511.08861.pdf,  MS-SSIM loss alone is a tricky energy,  and the mentioned paper also introduces a mixture L1 and MS-SSIM.  We do admit that there are some room left to study for the loss, and we plan addressing it further in the future works.

---

### Official Review · AnonReviewer1 · 2018-11-04
**Overall score 7**

**Rating:** 7
**Confidence:** 3

**Review:**

1. An ASPA coding method was proposed in this paper for lossy compression, which achieves the state-of-the-art performance on Kodak dataset and RAISE-1k dataset.

2. What is the difference on the architecture used in the proposed method and other compared methods? Since various numbers of layers or neurons lead to very big differences on the resulting performance.

3. Besides, it seems that the proposed method for compressing images includes more complex calculations. Is it faster or slower than others?

---

> ### Author Response · Authors · 2018-11-26
> **Thank you very much for the comments!**
>
> Thank you very much for the comments! We would like to respond to each one of them below, in order.
>
>
> "What is the difference on the architecture used in the proposed method and other compared methods? Since various numbers of layers or neurons lead to very big differences on the resulting performance."
>
> Practically, the most convoluted part of our algorithm is in our structure for the recursive construction of the latent variable z (Figure 11).  We have this structure for every k, which ranges from 1 to 10.   We do have to admit that our model is much deeper than the one used in Balle et al, which uses practically 7 layers for both encoder and decoder.
>
>
> "Besides, it seems that the proposed method for compressing images includes more
> complex calculations. Is it faster or slower than others? "
>
> When compared to the computation with fixed quantization width, we expect approximately 50% increase, because we are merely changing the parameters subject to the optimization from (mu, sig) to (mu, sig, q).  Also,  much of the computation in the quantization process can be parallelized (Figure 5), and in the light of that, the additional computational burden of concern shall not be much of a challenge.

---

### Official Review · AnonReviewer3 · 2018-11-06
**interesting approach; good MS-SSIM results; lacking insights / MSE evaluation**

**Rating:** 5
**Confidence:** 4

**Review:**

Adaptive Sample-space & Adaptive Probability (ASAP) lossy compressor is proposed. ASAP is based on neural networks.
ASAP jointly learns the quantization width and a corresponding adaptive quantization scheme. Some steps are similar to Nakanishi et al. 2018.
ASAP with the bpp x (1-MS-SSIM) x MSE loss improves in terms of MS-SSIM over Rippel & Bourdev 2017 and Nakanishi et al. 2018 on Kodak and RAISE-1k datasets.

The idea of jointly learning the quantization width and the adaptive quantization is interesting and the MS-SSIM results are good.

There is an inconsistency between section 2.2.2 and Fig. 7. I would expect in Fig. 7 also the result for the objective function from eq (6). In Fig. 7 could be added also a reference approach (such as BPG or a learned method).

By comparing Fig.7 and Fig.2  and Nakanishi et al., I suspect that the improvement of ASAP over Nakanishi et al., comes mainly from the change in the objective function and not from the proposed ASAP core?

The paper should include/discuss also the paper of
Balle et al., "Variational image compression with a scale hyperprior", ICLR 2018

The authors target only MS-SSIM, however it is unclear to me why.
According to Balle et al, ICLR 2018 and also to the recent CLIC challenge at CVPR18, learning for MS-SSIM or for PSNR / MSE leads to results that are not necessarily strongly correlated with the perceptual quality. MS-SSIM does not strongly correlate to the perceptual quality, while PSNR / MSE is a measure of fidelity / accuracy towards a ground truth.

I would like to see a comparison in PSNR / MSE terms with BPG and/or Balle et al., ICLR 2018.

I would like to see a discussion on the complexity, design, runtime, and memory requirements for the proposed approach in comparison with the other learned methods.

Also, it would be good to see more visual results, also for higher bpp.

---

> ### Author Response · Authors · 2018-11-26
> **Thank you very much for a thorough review!**
>
> Thank you very much for the comments and suggestions.   We reflected the suggestions on the revisions.
> Below, we would like to provide responses to the concerns raised:
>
>
> "There is an inconsistency between section 2.2.2 and Fig. 7. I would expect in Fig. 7 also the result for the objective
> function from eq (6). In Fig. 7 could be added also a reference approach (such as BPG or a learned method)."
>
> Thank you very much for pointing this typo. The caption in our original submission was wrong;   we meant BPP + lambda*(1 - MSSSIM) in all places we wrote BP + lambda * MSE.
> We fixed the legend in all graphs.
>
>
> "By comparing Fig.7 and Fig.2  and Nakanishi et al., I suspect that the improvement of ASAP over Nakanishi et al., comes mainly from the change in the objective function and not from the proposed ASAP core?"
>
> We shall first note that, as we state in the main article,  the result of our model in Fig.7 was produced by training the model over 0.1M, which is significantly less than the number of iterations used to produce the results for Fig 2 (0.3M).
> That being said, to make an assessment for this concern, we conducted an ablation study in which we compared the compression performance of our algorithm against those of the algorithm with fixed quantization width (all trained with the new objective function), and added a new figure illustrating the result.  On the model we trained for the benchmark study,  our ASAP was able to perform equally well as the compression with the ‘best’ fixed quantization width (best in terms of MS-SSIM).   For the assessment on the benchmark dataset, the benefit of our study turned out to be a relief from the burden of grid search.   We also conducted a separate ablation study with a smaller version of the model we used for a benchmark dataset.   For this second set of comparative experiments, we were able to confirm the benefit of the adaptive quantization size in terms of the rate-distortion tradeoff measured in MS-SSIM.  Adaptive width performed better than all choices of fixed quantization width.
>
>
> "The paper should include/discuss also the paper of Balle et al., "Variational image compression with a scale hyperprior", ICLR 2018"
> "The authors target only MS-SSIM, however it is unclear to me why. According to Balle et al, ICLR 2018 and also to the recent CLIC challenge at CVPR18,  learning for MS-SSIM or for PSNR / MSE leads to results that are not necessarily strongly correlated with the perceptual quality. MS-SSIM does not strongly correlate to the perceptual quality, while PSNR / MSE is a measure of fidelity / accuracy towards a ground truth.  I would like to see a comparison in PSNR / MSE terms with BPG and/or Balle et al., ICLR 2018."
>
> We mentioned the work of Balle et al in the script, and added their rate-distortion tradeoff curve in the figures. We also evaluated the performance of our method with PSNR scores,  and compare our method against Balle et al as well.   In terms of PSNR, our method(optimized for the novel loss)  was not able to perform better than Balle et al’s model optimized for PSNR.  In terms of MS-SSIM, our method performed better than Balle et al’s model optimized for MS-SSIM.
>
>
> "I would like to see a discussion on the complexity, design, runtime, and memory >requirements for the proposed approach in comparison with the other learned methods."
>
> Indeed, making the quantization width adaptive will increase computational cost to a certain extent.
> Roughty speaking, we do not expect much more than 50% increase in the sheer cost  (mu, sig) → (mu, sig, q).  Also,  as shown in Fig. 5, our computation admits parallel computing (the grids with same color can be computed in parallel). With the aid of GPU, this increase in computational burden shall not be much of a challenge.
>
>
> "Also, it would be good to see more visual results, also for higher bpp."
>
> We added the visual results for higher bpp.

---

### Meta-Review · Area_Chair1 · 2018-12-17
**Interesting work but important evaluation concerns remain**

**Confidence:** 4
**Recommendation:** Reject

**Metareview:**

This paper presents an interesting approach to image compression, as recognized by all reviewers. However, important concerns about evaluating the contribution remains: as noted by reviewers, evaluating the contribution requires disentangling what part of the improvement is due to the proposed approach and what part is due to the loss chosen and evaluation methods. While authors have done a valuable effort adding experiments to incorporate reviewers suggestions with ablation studies, it does not convincingly show that the proposed approach truly improves over existing ones like Balle et al. Authors are encouraged to strengthen their work for future submission by putting particular emphasis on those questions.